# Speed Matters: What to Prioritize in Optimization for Faster Websites

Christina Xilogianni, Filippos-Rafail Doukas, Ioannis C. Drivas * and Dimitrios Kouis

Information Management Research Lab, Department of Archival, Library and Information Studies, University of West Attica, 12243 Egaleo, Greece
* Correspondence: idrivas@uniwa.gr

**Abstract:** Website loading speed time matters when it comes to users' engagement and conversion rate optimization. The websites of libraries, archives, and museums (LAMs) are not an exception to this assumption. In this research paper, we propose a methodological assessment schema to evaluate the LAMs webpages' speed performance for a greater usability and navigability. The proposed methodology is composed of three different stages. First, the retrieval of the LAMs webpages' speed data is taking place. A sample of 121 cases of LAMs worldwide has been collected using the PageSpeed Insights tool of Google for their mobile and desktop performance. In the second stage, a statistical reliability and validity analysis takes place to propose a speed performance measurement system whose metrics express an internal cohesion and consistency. One step further, in the third stage, several predictive regression models are developed to discover which of the involved metrics impact mostly the total speed score of mobile or desktop versions of the examined webpages. The proposed methodology and the study's results could be helpful for LAMs administrators to set a data-driven framework of prioritization regarding the rectifications that need to be implemented for the optimized loading speed time of the webpages.

**Keywords:** website; webpage; loading time; speed; optimization; PageSpeed insights; libraries; archives; museums; LAM

## 1. Introduction

The digitalization of libraries, archives, and museums is in full swing and is growing. Nowadays, the ubiquitous presence of digital vehicles such as websites and social media platforms on the Web—shaped by the users, for the users—are more than capable of promoting the services of LAMs [1]. More than ever, a countermarch is taking place around the three pillars of a metaverse. There is a desire to *keep and protect*, to *experience and engage*, and *to analyze and optimize* the LAMs' content for a greater user experience with materials (libraries), holdings (archives), collections, and artefacts (museums).

Now and then, websites constitute one of the most prominent vehicles to promote the LAMs organizations' services and events without limitations. Each webpage within a whole website accepts full customization and content addition in quantity, aiming to disseminate the included information to stakeholders and interested parties with as much as optimized usability levels. From a practical point of view, websites allow LAMs to expand their visitation and audience navigation capabilities [2]. Compared to physical attendance, they also allow alternative types of interaction with the material, holdings, and collections [3]. Furthermore, LAMs websites can expand educational and cultural information, sometimes exceeding the physical volume of information within libraries, archives, and museum buildings [4]. As LAMs websites provide information to online visitors through their vast amount of educational and cultural content, they also cultivate both the prior-visit and the post-visit experience visit experience level [5].

Nevertheless, prior research approaches have already shed light on the low visibility and usability levels of LAMs websites on the Web [6,7]. Regarding the usability level, one of

the most fundamental characteristics within this aspect is the website loading speed [8–10]. Slow loading times are capable of dissatisfied users, decreasing in a significant manner their engagement with the provided website content in terms of the depth of exploration and the visit duration. A plurality of negative consequences has been discovered both in practical and research contexts due to the slow loading times of webpages. Such as lower user interactions [11], a reduced traffic acquisition [12], a higher cost per click in paid advertising context [13], and increase in bounce rates [14]. Controversially, a fast webpages loading speed time is aligned with an increased visit duration and pages per visit [15], higher organic search traffic and an increased number of sign-ups [16], a lower cost-per-click, and an increase in online revenues [17].

This paper proposes a methodological framework composed of three stages to measure the speed performance of the LAMs webpages. In the first stage, the retrieval of the speed performance data for mobile and desktop versions is taking place for 121 unique LAMs homepages worldwide. In the second stage, we evaluate the reliability and validity of the involved metrics aiming to develop a measurement system for assessing the speed performance of the examined LAMs webpages. After that, in the third stage, linear predictive regression models are developed. These models aim to understand which metrics have a higher impact than others on the total speed performance of the examined LAMs webpages. Understanding the impact of each metric will give the knowledge to LAMs administrators to support their prioritization strategy of "what to be optimized first" for a more excellent webpages loading time.

The rest of the study is unfolded into four parts. In the next second section, the importance of the websites' loading time and the related efforts to evaluate the speed performance are taking place. Additionally, the research gaps are discovered and discussed. The third section unfolds the methodology developed for a data retrieval, the tools used for this purpose, and the adopted statistical analysis methods. In the fourth section, the results are presented. Lastly, in the fifth section of the discussion, the practical, theoretical, and future implications are taking place.

## 2. Related Background

### 2.1. Importance of Websites Loading Time

It is a common phenomenon for a user to enter a website and, due to its slow loading speed time, abandon the website and return to the search engines' results pages to find other websites with faster loading content. Slow loading times cause a frustration to users, making them abandon webpages even if the content meets their informational needs [18]. In a general context, prior investigations showed that an increase in the loading time from one to three seconds could negatively impact the users' exploration, increasing in parallel the bounce rate by up to 32% [11]. Furthermore, BCC lost 10% of users for every additional second the website took to load its content [12]. Another recent study in Operations Research indicated that website visitors are most sensitive to website slowdowns at the checkout webpages and least sensitive to abandoning the content on homepages [14]. More specifically, the researchers indicated that a 10% increase in the load time is aligned with 1.7 percentage points of abandonments.

Speed matters when it comes to the users' experience in websites. When websites express a fast loading time, visitors become more engaged, and their interaction with the content increases in terms of the depth of exploration and visit duration [15]. High-performance websites, in terms of the speed component, retain users, exploring products and services further. For example, Pinterest increased the speed performance of specific landing pages for mobile devices, boosting organic search traffic, webpages per visit, and sign-up by up to 15% [16]. Another practical example comes from the COOK case, a growing company in the food industry. The conducted research found that the decrease in the website loading speed time by 850 milliseconds returned an increase in the conversion rate by up to 7%, pages per session by up to 10%, and a bounce rate reduction by up to 7% as well [19]. Regarding the paid search traffic performance optimization, Santander

increased the speed performance to the landing pages by 23%, increased the click-through rate by 11%, and decreased the cost-per-click by 22%, while the cost-per-conversion fell by 15% [20].

Speed also matters when it comes to online revenues. For Mobify, with the every 100 milliseconds decrease in the homepage and check-out webpage, session-based conversions increased from 1.11% to 1.53%. This decrease has been aligned with annual revenue of USD 380,000 up to USD 530,000 [21]. AutoAnything is another practical example. A faster loading time resulted in a 12–13% increase in online sales [22]. In addition, Furniture Village found that after decreasing the loading speed time of their website by 20%, there was an increase of 10% in the conversion rate for mobile devices, a 9% decrease in the bounce rate, and a 12% growth in online revenues coming from mobile devices [23].

### 2.2. Prior Efforts to Assess and Improve Speed Performance

One of the first attempts to highlight the speed performance and how it impacts the users' behavior within websites comes from Nah [24]. According to the findings of this study, the presence of feedback increases the Web users' tolerable waiting time. At the same time, the author indicated that the tolerable waiting time for an information retrieval is approximately 2 s. In the same year of research, Galleta et al. [25] indicated that the users' behavior in seeking information within websites began to flatten after waiting over 4 s. Another critical finding comes from Munyaradzi and colleagues [26]. The author suggested a rule of thumb regarding a website's size and its loading time, which should not exceed 8 s. A recent study investigating webpages' speed performance comes from the Indonesian region. The authors tried to harvest data through the GTMetrix tool to produce and present initial speed performance estimations from 61 journal websites and their adopted content management system (CMS) [27]. Their results indicated that although the Open Journal System articulates a better speed performance than other CMSs, most of the examined cases resulted in a moderate or poor speed performance.

Kaur et al. [28] proceeded into another study that measures the websites size and loading time using several web tools. The authors investigated up to twelve university websites from the Indian region. Several tools were used by the authors, such as Pingdom, to estimate the page size in bytes, Website Grader to estimate the overall performance, and server requests using the GTMetrix. They also used the PageSpeed Insights tool of Google (PSI), indicating that the examined webpages resulted in a load time ranging from 0.14 up to 10.81 s.

Regarding PageSpeed Insights, several efforts have already adopted the tool to estimate websites' performance. Similarly, Verma and Jaiswal [29] used PSI to examine the top-thirty medical universities' websites in the Indian region. They proceeded into an initial holistic performance estimation for the 30 examined webpages using the two primary metrics of PSI: the PageSpeed Insights score for mobile and desktop. In almost the same manner, one year later, Patel and Vyas [30] tried to examine the performance of the websites' of the open universities in India once more. The researchers used several web tools to retrieve technical performance data about their examined websites cases. Among them, PSI has been used to estimate the speed performance for mobile and desktop. In addition, a detailed comparison took place regarding the higher performance per examined website. Nevertheless, it was out of the researchers' scope to include suggestions for improving the speed performance.

Another research contribution using PSI comes from Amjad et al. [31]. The authors tried to harvest the performance of ten e-commerce websites in Bangladesh using multiple parameters. Such as the fully loaded time of requests, first CPU idle, speed index, start render, fully loaded time of the content, document complete time, last painted hero, first contentful paint, and the first byte. PSI has been used to retrieve data for some of these parameters: the speed index, first CPU idle, and first contentful paint. Likewise to Patel and Vyas [30], Amjad and colleagues [31] proceeded to compare the results among the

examined e-commerce websites. However, for all the authors above, it was out of their research scope to suggest practical implications for improving the speed performance.

Bocchi et al. [32] used three different web tools to examine the quality of experience of web users while also trying to find similarities in the metrics results among the three tools. They used PSI, Yahoo's YSlow, and Show Slow, developed by Chernyshev [33]. They retrieved data are from the top 100 websites as Alexa listed them to discover possible correlations between the above speed performance tools. The results did not show statistically significant correlations between the PSI and the other tools. However, the authors proposed other metrics that focused exclusively on simplifying the performance measurement systems of the website speed.

In another study coming from the LAMs' context, Krstić and Masliković [34] used PSI to examine the speed performance of the libraries, archives, and museum institutions of the Serbia region. The authors extracted a low up to moderate the speed performance for the examined LAMs. More specifically, national libraries resulted in a score of 52/100, archives up to 55/100, and national museums up to 51/100. However, there was no explicit indication if these results correspond to the mobile or desktop speed performance scores. Moreover, Redkina [35] examined six domains of national libraries regarding their speed performance in mobile and desktop devices through the PSI. Generally, all the results indicated a difference in the rates as the mobile performance ranged from 8 to 16 units less than the desktop estimations.

However, based on the current research efforts, several gaps are derived. Most of these studies do not investigate the PSI metrics in depth, but they consider only the two primary metrics: the total speed performance on mobile and desktop devices. In addition, except for one study by Bocchi et al. [32], the rest investigate only a limited number of cases. Therefore, further effort is needed to examine a larger number of webpages and the impact of each PSI metric on their loading speed time. This effort will also increase the generalizability of the results regarding the code elements that impact the webpage loading time. Moreover, it is noted that apart from Krstić and Masliković [34] and Redkina [35], there is no other prior research effort that involves a set of metrics to examine the LAMs webpages' speed performance.

Another research gap is related to the efforts of transforming the derived technical analytics into valuable and practical insights that administrators could utilize to improve the webpages' speed performance. More specifically, there is a need to propose a methodology capable of prioritizing which of the speed metrics needs rectification first, aiming to reduce the speed loading time while setting priorities within the context of the requirements of the engineering strategy [36]. In other words, we assume that some metrics might impact on a higher level to reduce the webpages' loading time, compared to others, on desktop and mobile devices. By extracting this kind of information, LAMs administrators could support, in a well-informed way, what to prioritize for optimization and why in specific webpages while combining both analytical thinking and prior knowledge and experience [37].

The loading time of webpages concerning their impact on the total performance of a landing page is another aspect that needs to be highlighted. More specifically, LAMs administrators often try to develop campaigns for promoting services or organizational events, creating landing pages where potential visitors will get information about these services or events. However, visitors will get frustrated if the landing page does not express a sufficient loading speed time. For example, in the case of paid search advertising campaigns, if the usability of the landing page is poor in terms of the loading time, the cost-per-click will be increased, as the quality of a landing page impacts the ads' performance [13,38]. Therefore, it should be beneficial for LAMs administrators to get an in-depth overview of the metrics that impact most webpages' speed performance, incorporating this knowledge into the landing pages' design and development phase.

Lastly, another drawback is related to the existence of a speed performance measurement system that estimates the loading speed time of a webpage expressing reliability, internal consistency, and validity in terms of its involved metrics. Covering this gap will

help LAMs administrators rely on a speed performance measurement system that, if re-used, will return in similar results and indications for improvement [39]. One step further, there is a need to deploy the proposed webpage speed performance measurement system in a large-scale amount of LAMs cases from all over the world. This will increase the generalization of the results and the reliability that the proposed performance measurement system stands with a sufficient internal consistency and validity. In other words, the more the LAMs webpages cases are examined, the more the results will depict the overall picture of their speed performance reliably and validly.

In Table 1, we summarize the research gaps, while in the next section, a methodology is proposed to cover these gaps.

**Table 1.** Derived research issues and reflections in webpages speed performance in LAMs context.

| Research Issues | Contribution Needed |
|---|---|
| Most studies investigate webpages' speed performance involving a limited number of cases. | Further research is needed to increase the number of cases improving the generalization of results. |
| There is a need to explore in-depth the webpages' speed performance tools and only their general indicators that reflect only the total performance of speed in mobile or desktop devices. | An in-depth investigation is needed to examine a plurality of metrics included in the webpages' speed performance tools, aiming to understand which metrics impact most to the increase or decrease of the total speed performance. |
| Limited efforts in the LAMs context focus on prioritizing what rectifications need to be done, among others, to increase webpage's speed performance. | A research effort is needed to develop predictive models that estimate which metrics impact most to the total webpages speed performance, improving in this way requirements prioritization strategy. |
| In continuation of requirements prioritization need for further research, it should be helpful to examine which technical aspects impact landing pages speed performance to improve users' engagement when they arrived. | A research effort is needed to develop predictive models that estimate which metrics impact most to the total webpages speed performance, improving in this way requirements prioritization strategy. |
| There is a lack of a reliable and valid speed performance measurement system that expresses sufficient internal consistency regarding its involved metrics. | There is a need to develop a reliable and valid speed performance measurement system for webpages that other LAMs administrators could potentially use for their websites. |

## 3. Materials and Methods

### 3.1. Data Collection and Sample

As previously mentioned in the introductory part of this research, we shed light on the technical aspects that impact the most on the loading speed performance of LAMs' webpages. From an information systems point of view, this works both at the micro and macro level [40]. In terms of micro-level optimization, discovering the elements that mostly impact on a single webpage's loading time, constitutes one of the first steps to increasing its speed. Regarding the macro-level, as each one of the "parts" are improved (single webpages within a website), then eventually, the "whole" is improved as well (the website).

For gathering data about the webpages' speed performance both on desktop and mobile devices, we used Google PageSpeed Insights. We justify our selection among the other tools that estimate the speed performance for two main reasons. First, through a micro-level perspective, after extracting the results regarding the specific metrics (see Table 2), PSI suggests to developers specific rectifications in the HTML, JavaScript, and CSS elements of the code or even a reduction in the size of images. These suggestions aligned with a precise reduction in seconds or milliseconds for each line of the website code that could be rectified. Additionally, second, this performance tool differentiates webpage speed performance into mobile and desktop reports. This fact gives LAMs websites administrators a complete understanding of how particular webpages technically behave both for the users coming from mobile or desktop devices.

We gathered speed data for 121 LAMs websites from all over the world. Each test per website extracted two reports, the mobile and desktop performance. Six metrics were presented for every performance report, and their data were retrieved, making a total of 12 metrics (6 on mobile and 6 on desktop). Both for mobile and desktop, the 6 metrics were the same. However, they extracted different results per version. PageSpeed Insights allows websites administrators to estimate the speed performance of a webpage and not the whole website, such as other tools that have been used in the prior research context of the website's performance measurement and optimization [41]. Therefore, we estimated the speed performance of the unique webpages of the examined LAMs websites.

**Table 2.** PageSpeed Insights involved metrics explained.

| Page Speed Insights Metrics | Metrics Explanation |
|---|---|
| First contentful paint (FCP) | It is calculated in seconds. The FCP estimates how long it takes the browser to load the first piece of Document Object Model (DOM), after the user enters the webpage. Based on Google PageSpeed Insights documentation, 0–1.8 s is a good FCP, 1.8–3.00 s moderate, and over 3 s slow. |
| Speed index (SI) | It is calculated in seconds. The speed index metric estimates how quickly the content is displayed during webpage loading. According to Google documentation, speed index between 0 and 3.4 s is characterized as fast loading, 3.4 to 5.8 moderate, and over 5.8 as slow. |
| Largest contentful paint (LCP) | Just like the prior metrics, it is calculated in seconds. The LCP metric reports the render of time that the most extensive image or text block needs to be loaded on the user's screen compared to the time when the first page started to load. Less than 2.5 s is a good LCP performance. Between 2.6 and 4 s is a moderate one, while over 4.1 is a poor LCP performance. |
| Time to interactive (TtI) | It is calculated in seconds. Sometimes we try to interact with some elements of a webpage (such as buttons, open accordions, etc.), but they are not ready for it, which probably frustrates users. The TtI metric measure how long it takes for a webpage to be fully interactive to the user. 0 up to 3.8 s characterized as fast TiI, 3.9 up to 7.3 as moderate, and over 7.3 as slow. |
| Total blocking time (TBT) | It is measured in milliseconds. It estimates the total amount of a webpage is blocked before it responds to the user, such as after mouse clicks, screen, or keyboard taps. Milliseconds less than 200 are characterized as fast, 200 up to 600 as moderate, and over 600 as slow. |
| Cumulative layout shift (CLS) | It is calculated in seconds. The CLS estimates the total times layouts of a webpage are shifting before their final solidification. For example, users enter the webpage, and suddenly the text is moving, or the link they want to click changes its position. This is a layout shift. The more the layouts shift, the higher the number of seconds a user needs to browse a webpage. Based on Google PageSpeed Insights documentation, 0.1 s is characterized as good, 0.1 up to 0.25 s is moderate, and over 0.25 as slow. |
| PageSpeed Insights score (PSI) | It is estimated using all the metrics mentioned above with a scale ranging from 0 to 100. The higher the number, the better the webpage speed performance. |

The homepage of each one of the 121 cases has been selected. We selected the homepage per website exclusively as it constitutes one of the vital neurons of each domain, offering the opportunity for an organization to give a holistic rundown of its brand identity, services, and provision of products to incoming visitors [42]. We believe that LAMs are not an exception to the assumption of Geissler et al.; thus, we deployed PageSpeed Insights to the homepage of each of the 121 cases. Descriptive statistics have been used to estimate the initial performance of the examined webpages using the mean, std deviation, min, max, and skewness. The latter estimates the tendency of metrics and if the sample per metric is closer to the minimum or the maximum values. In addition, Shapiro–Wilk has been used to test the normal distribution of the metrics as one of the most potent normality tests [43].

In Table 2, we explain the metrics, while Tables 3 and 4 present a sample of the data. This will help the study readers understand the research data sample and how it was analyzed.

**Table 3.** Sample of a dataset regarding a random selection of three different LAMs and their PSI performance in mobile devices.

| Webpage Examined | PageSpeed Insights Total Score | First Contentful Paint | Speed Index | Largest Contentful Paint | Time to Interactive | Total Blocking Time | Cumulative Layout Shift |
|---|---|---|---|---|---|---|---|
| museunacional.cat | 30 | 13.2 | 21.3 | 17.8 | 20 | 260 | 0.36 |
| artgallery.nsw.gov.au | 44 | 3.5 | 5.3 | 6.4 | 9.3 | 630 | 0.003 |
| nfsa.gov.au | 27 | 4.4 | 10.09 | 9.9 | 9.4 | 370 | 0.846 |

**Table 4.** Sample of a dataset regarding a random selection of three different LAMs and their PSI performance in desktop devices.

| Webpage Examined | PageSpeed Insights Total Score | First Contentful Paint | Speed Index | Largest Contentful Paint | Time to Interactive | Total Blocking Time | Cumulative Layout Shift |
|---|---|---|---|---|---|---|---|
| nelson-atkins.org | 65 | 1 | 3.1 | 7.2 | 2.2 | 90 | 0.052 |
| high.org | 79 | 0.8 | 2 | 2.6 | 2.9 | 0 | 0.004 |
| nasjonalmuseet.no | 85 | 0.9 | 1 | 2.5 | 1.8 | 20 | 0.014 |

### 3.2. Validity and Reliability Assessment

In this study, an effort is made to examine whether the proposed metrics, both for mobile and desktop, express a statistical reliability, cohesion, and consistency. To do this, we deployed several different tests. The reason behind the validity and reliability assessment

of the involved metrics is related to the confirmation that they could be used potentially for other websites not only in the LAMs sector but in different domains of interest as well. In other words, if a construct expresses an internal discriminant validity, cohesion, and consistency, then probabilities are increased to replicate this performance measurement system in a similar problem, expecting similar reliability results [44,45]. For the assessment of the reliability, we deployed, at the initial level, an exploratory factor analysis (EFA) using the indicators of Kaiser–Meyer–Olkin, $\chi 2$ and Bartlett's test of Sphericity to measure the goodness of fit among the two constructs of the metrics [46]. For the factor loadings extraction, a KMO threshold of exclusion has been set to 0.5 [47]. At the early stage of building a statistically reliable assessment schema, exploratory rather than confirmatory factor analysis is preferable [48]. That is, EFA is selected to determine the factor structure, while CFA is selected to verify the nature of the proposed factor structure [49].

One layer down, and to ensure that the proposed involved metrics per construct express a statistical reliability and discriminant validity, a confirmatory factor analysis (CFA) is performed. In CFA, the researcher is not concerned exclusively with discovering or disclosing new constructs, but instead quantifying and confirming a priori or pre-determined and pre-convinced structures of the possible interrelationships between a set of considered measures [50]. The KMO, the extracted $\chi 2$ from EFA, Bartlett's test, comparative fit index (CFI), root mean square error of approximation (RMSEA), McDonald's $\omega$, Cronbach's $\alpha$, and Guttman's $\lambda$-2 tests were used as indicators to measure the reliability and validity levels of the proposed constructs, that is the mobile and desktop speed performance. The CFI ranges from 0 to 1, while the higher the value, the better the model's fit. A rule of thumb indicates a CFI value starting from 0.90 or greater [51]. The RMSEA indicates the residual in the proposed model, and the values range from 0 to 1, as the smaller the value, the greater the model fit. An acceptable model fit indicates values of 0.06 or less [52].

McDonald's $\omega$ was used to estimate the strength's association among the involved variables within the two constructs [53] with values ranging from 0 to 1. The greater the association, the higher the value. Cronbach's $\alpha$ was used as one of the most common acceptance estimators of the proposed two constructs, while the closer the values to 1, the better the level of acceptance. Guttman's $\lambda$-2 worked supportively to Cronbach's $\alpha$ for estimating the variance trustworthiness among the examined metrics in each construct [54, 55]. Lastly, Guttman's $\lambda6$ indicator has also been used as it estimates the variance in which each metric could be involved in the development of linear regression models [56].

### 3.3. Predictive Regression Models

After the reliability and validity analysis to estimate the internal cohesion and consistency of the proposed two constructs, an effort is given to developing predictive linear regression models that can specify which metric impacts on a higher level to the degree of change in mobile and desktop webpages' speed performance. In other words, we try to understand which of the metrics impact on a higher level to the increase or decrease in the total Pagespeed Insights score in the mobile and desktop performance of the examined 121 LAMs homepages. This extracted knowledge will reinforce the decision-making process of the LAMs administrators to prioritize optimization steps while setting forth the improvement of the metrics that impact on a higher level to the total website speed performance compared to others with a lower level of impact. The predictive regression models fit and the predictive capability is designated through $R^2$, adjusted $R^2$, *p* values, and F. For the latter, the higher the F value, the better and stronger the predictive capability [57].

In the next section, we present the results of our study, while in the following Figure 1, the overall methodology developed is depicted.

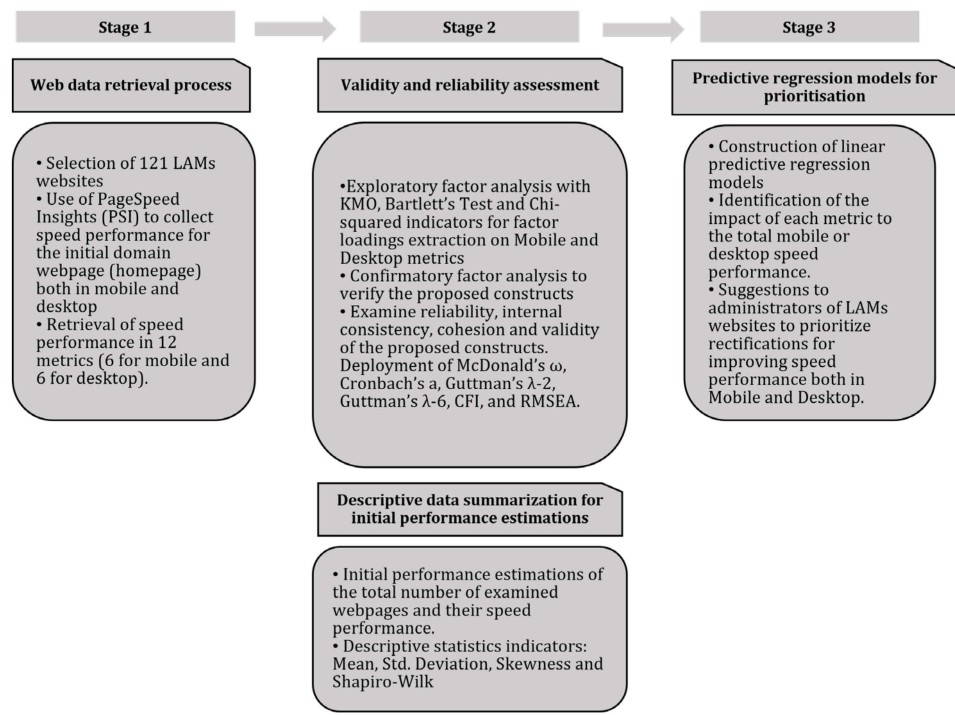

**Figure 1.** Proposed methodology to understanding webpages speed performance and prioritize rectifications through the development of predictive models.

## 4. Results

### 4.1. Reliability Analysis Results for Mobile and Desktop

In the subsequent Tables 5 and 6, the results from the reliability analysis are taking place. Regarding the variable loading per construct, both mobile and desktop indicated sufficient values, while none of them ranged below the recommended level of exclusion (0.500). For the mobile construct, variables loadings ranged from 0.567 (cumulative layout shift) up to 0.810 (total blocking time). For the desktop construct, variable loadings ranged from 0.529 (cumulative layout shift) to 0.836 (largest contentful paint).

**Table 5.** Exploratory factor analysis results and loadings for Mobile and Desktop Metrics of PageSpeed Insights.

| PageSpeed Insights Mobile | | PageSpeed Insights Desktop | |
|---|---|---|---|
| **Variables** | **Variable Loadings** | **Variables** | **Variable Loadings** |
| First contentful paint | 0.614 | First contentful paint | 0.626 |
| Speed index | 0.723 | Speed index | 0.750 |
| Largest contentful paint | 0.883 | Largest contentful paint | 0.836 |
| Time to interactive | 0.747 | Time to interactive | 0.760 |
| Total blocking time | 0.810 | Total blocking time | 0.655 |
| Cumulative layout shift | 0.567 | Cumulative layout shift | 0.529 |

**Table 6.** Internal consistency and validity of the two constructs, PageSpeed Insights mobile and desktop.

| Constructs | KMO and Overall Sampling of Adequacy | EFA Bartlett's Sphericity *p* Value | EFA χ2 *p* Value | CFI | RMSEA | McDonald's ω | Cronbach's α | Guttman's λ-2 | Guttman's λ-6 |
|---|---|---|---|---|---|---|---|---|---|
| PageSpeed Insights mobile | 0.758 | <0.001 | <0.001 | 0.921 | 0.014 | 0.679 | 0.752 | 0.678 | 0.807 |
| PageSpeed Insights desktop | 0.709 | <0.001 | <0.001 | 0.917 | 0.023 | 0.642 | 0.742 | 0.711 | 0.797 |

In terms of the remainder of the statistical tests that indicate the level of reliability and validity per construct, KMO indicated sufficient measurements of sampling adequacy both for mobile (0.758) and desktop (0.709). Furthermore, the comparative fit index indicator extracted values above the recommended tolerance of greater than 0.90 according to [51] being both on mobile (0.921) and desktop (0.917). Regarding the recommended tolerance value in the RMSEA indicator, the mobile construct extracted up to 0.014 and desktop up to 0.023. Therefore, the RMSEA values are acceptable as they are below the threshold of 0.06 [52] both for mobile and desktop constructs.

Furthermore, the construct reliability fit indicators of McDonald's ω, Cronbach's α, Guttman's λ-2, and λ-6 ranged in sufficient and acceptable values. More specifically, for mobile, their values ranged from 0.678 (Guttman's λ-2) up to 0.807 (Guttman's λ-6); as for desktop, these values ranged from 0.642 (McDonald's ω) up to 0.797. In this respect, the results support our assumption that the proposed two constructs express a statistical reliability and validity. This fact works as a solid steppingstone to trust and replicates the potential webpages' speed performance based on these two constructs, expecting similar reliability levels [44,45].

*4.2. Descriptive Statistics for Summarizations and Initial Performance Estimations*

In Tables 7 and 8, we present the descriptive statistics for all the metrics, including the PageSpeed Insights total score. Moreover, in Table 9, the difference between the mobile and desktop performance metrics is presented, helping the study readers to understand the examined LAMs webpages performance at an initial level.

**Table 7.** Descriptive statistics results for mobile PageSpeed Insights metrics.

| | Pagespeed Insights Total-Mobile | First Contentful Paint | Speed Index | Largest Contentful Paint | Time to Interactive | Total Blocking Time | Cumulative Layout Shift |
|---|---|---|---|---|---|---|---|
| Mean | 38.656 | 4.819 | 12.167 | 13.554 | 15.236 | 701.745 | 0.231 |
| Std. deviation | 18.114 | 3.085 | 10.098 | 11.791 | 13.337 | 1384.725 | 0.305 |
| Skewness | 0.498 | 2.501 | 3.066 | 3.442 | 4.484 | 6.767 | 1.849 |
| Shapiro–Wilk | 0.962 | 0.751 | 0.715 | 0.682 | 0.630 | 0.407 | 0.760 |
| Shapiro–Wilk *p* value | 0.121 | 0.119 | 0.075 | 0.089 | 0.111 | 0.063 | 0.119 |
| Minimum | 3.00 | 1.20 | 2.80 | 2.10 | 3.00 | 0.00 | 0.00 |
| Maximum | 90.00 | 21.40 | 76.40 | 93.00 | 117.90 | 13,250.0 | 1.516 |

*N = 121 | chosen alpha level of Shapiro–Wilk p value at >0.05.*

**Table 8.** Descriptive statistics results for Desktop PageSpeed Insights Metrics.

| | PageSpeed Insights Total-Desktop | First Contentful Paint | Speed Index | Largest Contentful Paint | Time to Interactive | Total Blocking Time | Cumulative Layout Shift |
|---|---|---|---|---|---|---|---|
| Mean | 69.615 | 1.268 | 3.456 | 3.374 | 2.912 | 99.788 | 0.132 |
| Std. deviation | 17.776 | 0.793 | 2.585 | 2.266 | 2.720 | 272.983 | 0.198 |
| Skewness | −0.581 | 2.300 | 1.904 | 2.213 | 4.109 | 7.094 | 2.172 |
| Shapiro–Wilk | 0.966 | 0.757 | 0.806 | 0.815 | 0.657 | 0.348 | 0.695 |
| Shapiro–Wilk *p* value | 0.171 | 0.141 | 0.059 | 0.071 | 0.074 | 0.063 | 0.129 |
| Minimum | 18.000 | 0.400 | 0.800 | 0.700 | 0.600 | 0.000 | 0.000 |
| Maximum | 99.000 | 4.800 | 14.300 | 16.400 | 23.200 | 2600.000 | 1.042 |

*N = 121 | chosen alpha level of Shapiro–Wilk p value at >0.05.*

**Table 9.** Difference between mobile and desktop scores per metric.

| | Mobile | Desktop | Difference Between Metrics |
|---|---|---|---|
| PageSpeed Insights total | 38.656 | 69.615 | −30.959 |
| First contentful paint | 4.819 | 1.268 | 3.551 |
| Speed index | 12.167 | 3.456 | 8.711 |
| Largest contentful paint | 13.554 | 3.374 | 10.18 |
| Time to interactive | 15.236 | 2.912 | 12.324 |
| Total blocking time | 701.745 | 99.788 | 601.957 |
| Cumulative layout shift | 0.231 | 0.132 | 0.099 |

In terms of the PageSpeed Insights total score values, it is noted that the mobile mean value resulted in 38.656, comparatively with the desktop that extracted a mean value of 69.615. That is a total difference of 30.9 units between the two metrics. Regarding the first contentful paint in the mobile version of the examined LAMs websites, a mean value of

4.819 was extracted, while for the desktop, the mean value for the same metric was 1.268. Similarly, the speed index, the largest contentful paint, and the time to interactive extracted lower mean values in mobile metrics rather than desktop. For the metric total blocking time, the mobile performance indicated a mean value of 701.745 milliseconds, while for the desktop, the mean value of this metric ranged significantly to a lower level with 99.788 milliseconds. Moreover, a difference of 0.099 was observed in the cumulative layout shift metric as the mean value for mobile was 0.231 and for desktop was 0.132.

Regarding the normality of the retrieved metrics, all of them, both for mobile and desktop, resulted sufficient Shapiro–Wilk indicators and *p* values greater than the chosen alpha level of 0.05, indicating, in this way, that they did not show evidence of non-normality to their distribution. In particular, for mobile, the Shapiro–Wilk ranged from 0.407 (total blocking time) to 0.962 (Pagespeed Insights total—mobile). For desktop metrics, the Shapiro–Wilk ranged from 0.348 (total blocking time) to 0.966 (PageSpeed Insights total—desktop).

Lastly, it is observed that for most of the metrics, the skewness indicator pointed out positive values for mobile and desktop. This result means that the examined metrics tend to direct to the maximum values rather than the minimum ones. In other words, as most of the metrics of the PSI was calculated in seconds and milliseconds, the more these values are, the slower is the speed performance of the examined LAMs websites. The only exception is the PageSpeed Insights total—desktop metric, where skewness resulted in a negative value of $-0.581$. Additionally, indeed, the mean value of this metric tended to be closer to the maximum value than the minimum one.

### 4.3. Predictive Regression Models

In Tables 10 and 11, we present the regression equation outputs. The scope of these predictive models is to indicate which metrics impact on a higher level to the total speed performance of a webpage. This will help drastically to prioritize the rectifications for optimizing the speed performance of the examined LAMs homepages. Considering the *p* values of all the linear regression outputs, most of them articulated a high statistical significance of $p < 0.001$. These results support the assumption that the proposed predictive models can reject the null hypothesis, that is, to extract regression coefficients equal to zero values, which means that the developed models lag behind a predictive discriminant capability [57]. Based on the results of this study, all the models reject the null hypothesis and express a sufficient discriminant capability while taking into consideration both *p* and F values, respectively.

**Table 10.** Regression equation output of mobile PageSpeed Insights metrics and their potential predicted impact on PageSpeed Insights total—mobile.

| Variable | Coefficient | $R^2$ | Adjusted $R^2$ | F | *p* Value |
|---|---|---|---|---|---|
| Constant (PageSpeed Insights total—mobile) | 47.782 | 0.104 | 0.097 | 13.927 | 0.014 |
| First contentful paint | 1.894 | | | | |
| Constant | 48.786 | 0.215 | 0.209 | 32.949 | <0.001 |
| Speed index | 0.833 | | | | |
| Constant | 48.585 | 0.227 | 0.221 | 35.320 | <0.001 |
| Largest contentful paint | 0.733 | | | | |
| Constant | 48.581 | 0.230 | 0.224 | 35.851 | <0.001 |
| Time to interactive | 0.651 | | | | |
| Constant | 41.946 | 0.128 | 0.121 | 17.693 | <0.001 |
| Total blocking time | 0.165 | | | | |
| Constant | 44.226 | 0.166 | 0.159 | 23.827 | <0.001 |
| Cumulative layout shift | 1.212 | | | | |

**Table 11.** Regression equation output of desktop PageSpeed Insights metrics and their potential predicted impact on PageSpeed Insights total—desktop.

| Variable | Coefficient | $R^2$ | Adjusted $R^2$ | F | *p* Value |
|---|---|---|---|---|---|
| Constant (PageSpeed Insights total—desktop) | 83.715 | 0.246 | 0.240 | 39.127 | <0.001 |
| First contentful paint | 5.223 | | | | |
| Constant | 85.820 | 0.465 | 0.461 | 104.299 | <0.001 |
| Speed index | 4.689 | | | | |
| Constant | 89.103 | 0.542 | 0.539 | 142.214 | <0.001 |
| Largest contentful paint | 5.776 | | | | |
| Constant | 82.786 | 0.479 | 0.475 | 110.350 | <0.001 |
| Time to interactive | 4.522 | | | | |
| Constant | 72.652 | 0.218 | 0.212 | 33.548 | <0.001 |
| Total blocking time | 0.030 | | | | |
| Constant | 73.530 | 0.109 | 0.109 | 14.670 | 0.004 |
| Cumulative layout shift | 4.112 | | | | |

To begin with, in Table 10, the potential predicted value of a change in the metric PageSpeed Insights total—mobile (constant) (herein PSI total-mobile) is presented. A high statistical significance was observed with a *p* value of 0.014 and an $R^2$ of 0.104 between the PSI total—mobile and first contentful paint. More specifically, for each unit that first contentful paint is optimized, the PSI total mobile metric is expected to increase by 1.894. The speed index ($R^2$ of 0.215 and *p* < 0.001) increased the PSI total—mobile by 0.833. Furthermore, the second higher value of the predicted change after the first contentful paint metric is expected in the cumulative layout shift value by 1.212 ($R^2$ of 0.166 and *p* < 0.001). In terms of the lowest predictive value of change, PSI total—mobile is expected to increase by 0.165 for each percentage unit of the total blocking time ($R^2$ of 0.128 and *p* < 0.001).

Continuing with the regression equation outputs of the potential predicted value of change in the metric PageSpeed Insights total—desktop (constant) (herein PSI total-desktop), several differentiations are observed compared to the regression results for mobile. More specifically, in almost all the metrics, the PSI total—desktop indicated higher values of the predicted change. For example, for each one second of improvement in the first contentful paint, the PSI total—desktop metric could be increased by 5.233 ($R^2$ of 0.246 and *p* value of < 0.001). Compared with the mobile, the predictive value of change reached up to 1.894. The highest predictive value of change has been observed in the largest contentful paint ($R^2$ of 0.542 and *p* value of < 0.001). That is, for each one-second improvement in this metric, the PSI total—desktop metric could be increased by up to 5.776.

It is noted that among the other desktop metrics, the largest contentful paint indicates the highest predictive power as a linear predictive model with an $R^2$ of 0.542 and an F value of 142.214. The lowest predictive value of change for the desktop has been extracted to the total blocking time metric. For the latter, in every millisecond of improvement, the total PSI—desktop could be increased by 0.030 ($R^2$ of 0.218 and *p* value < 0.001).

The following two bar charts in Figures 2 and 3 visualize the extracted predicted value of change that is expected both for mobile and desktop for each one of the six common metrics. This will help practically administrator LAMs to understand in an easy and meaningful way which are the technical components that need to be prioritized for optimizing the speed performance of their webpages.

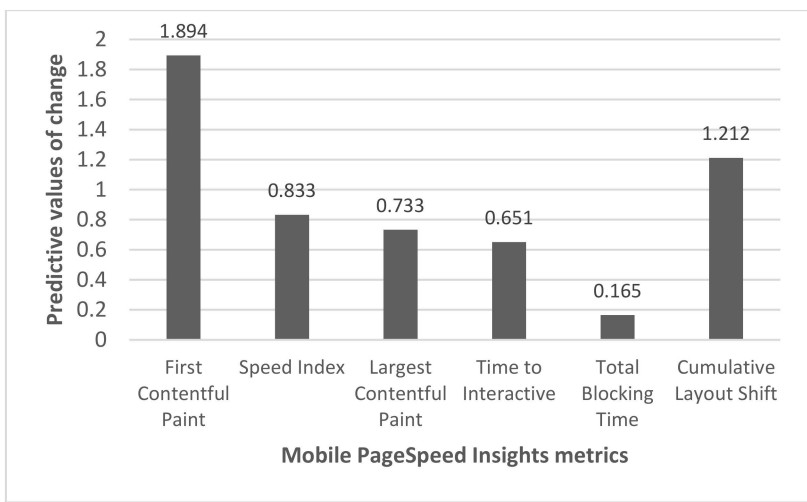

**Figure 2.** Predictive impact of change of total PageSpeed Insights performance per metric for the examined LAMs websites and their mobile versions.

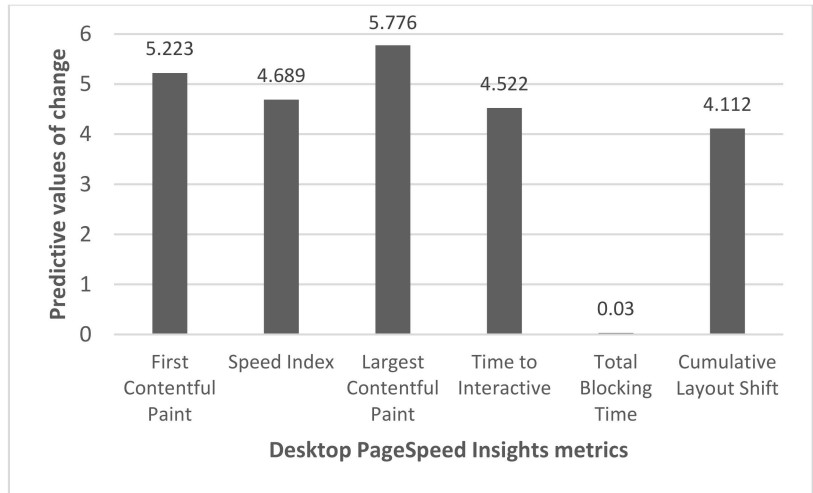

**Figure 3.** Predictive impact of change of total PageSpeed Insights performance per metric for the examined LAMs websites and their desktop versions.

In the following section, we unfold the practical–managerial and theoretical contributions of this study.

## 5. Discussion

In this study, we examined the factors that mostly impact the speed performance of the LAMs' websites. More specifically, the PageSpeed Insights of Google has been used to assess the speed performance both for mobile and desktop through the involvement of 12 metrics for 121 unique cases. The motivation behind this is related to the effort of understanding which components impact most on the loading speed time of the LAMs' homepages, aiming to increase the user usability and navigation experience. In the following sub-sections, we highlight the study's contribution from both a practical and theoretical perspective.

### 5.1. Webpages Speed Performance Comparisons

In this study, we used PSI to extract the homepages speed performance of 121 LAMs worldwide, both for mobile and desktop devices. Several significant findings were discovered by comparing the results among the two types of webpages' speed performance. First, the desktop version of the homepages resulted in a greater performance than the mobile version. For example, the PageSpeed Insights total for mobile resulted in a mean value

of 38.65/100, while on the other hand, the desktop mean values ranged up to 69.61/100, observing a difference of more than 30 units.

First contentful paint indicated a mean value of up to 4.81 s for the mobile versions of the examined homepages, while the desktop indicated a mean value of 1.26 s (a difference approximately of 3.55 s between them). The speed index, largest contentful paint, and time to interactive expressed lower mean values of seconds on desktop compared to mobile. Moreover, the total blocking time indicates by far a higher mean value in mobile (701.74 milliseconds) compared to desktop (99.78 milliseconds), while a cumulative layout shift resulted in 0.231 s for mobile, while for desktop it was 0.132. This difference in all the metrics clearly indicates the lower performance of the examined homepages of LAMs in mobile versions.

Similar differences were also observed in the values of predicted changes through the regression equation outputs. More specifically, it is notable that the mobile speed performance extracted lower values of change compared to desktop. For example, for each unit that the first contentful paint increased by, the total mobile performance increased by up to 1.894. On the other hand, the same metric for desktop extracts up to 5.223. Similar predicted values of change are observed in the rest of the metrics as the desktop version of the LAMs' homepages are more in a favorable position to increase their loading speed performance. The only exception is observed in the total blocking time, where for every millisecond decrease for the mobile devices, the total mobile performance is increased by 0.165, compared to the desktop, which articulates a change of 0.03.

However, the mobile speed performance matters. Without a doubt, a webpage that expresses a low level of mobile speed performance increases the probability of potentially dissatisfying the navigation experience of mobile users, and also decreases the depth of exploration both in duration and pageviews. In addition, search engines consider the reduced mobile-friendliness of such webpages feeding algorithms to place lower rankings in the search results, hence decreasing organic search traffic [58]. Controversially, adaptive mobile experiences in webpages return friendly and high-speed browsing. In this respect, the results of this study ring the bell to the administrators to optimize the library, archives, and museum homepages' speed performance, expecting a better user engagement with the content and lower bounce rates.

### 5.2. Practical-Managerial Implications

From an information system thinking point of view, without initial performance estimations, any kind of optimization within the systems and its parts is at least difficult to be implemented [40]. Continuously, for each kind of assessment, a performance measurement framework is needed in order to estimate the initial competency of what is measured within a system. In this study, we explored the PSI and its involved metrics to examine its reliability and internal consistency as a tool to assess the webpages' speed performance. Several statistical analyses have examined the tool's internal consistency, cohesion, and discriminant capability. Results indicated that the proposed schema expresses reliability and validity (Tables 5 and 6). The reason behind this analysis was to develop and suggest a reliable performance measurement system that could be used by websites administrators aiming to increase the speed performance.

One step forward, several regression models have been developed, shedding the light to practitioners on what to prioritize within webpage optimization and why. Through these regression models that have been developed, some metrics extracted higher predicted values of change compared to others. Of course, these predictive values of change correspond to the examined sample, and probably different webpages selections will extract some other metric (of these six) to express the highest predictive value of change. In any case, as the proposed webpages' speed performance measurement system expressed a statistical reliability and validity, the LAMs administrators can utilize the proposed assessment schema to evaluate the webpages of their selection.

This data-driven analytical thinking could also be utilized to assess the homepages' speed performance and landing pages. That is, (*a*) use the PSI tool in specific landing pages that contain services or announcements of events, (*b*) retrieve data from the proposed metrics for these specific landing pages, (*c*) develop linear predictive regression models that prioritize which metric(s) need to improve first, and after that, (*d*) proceeded into code rectifications to reduce the loading time. Actually, in terms of a paid search advertising optimization, even if a campaign is overhauled with quality in ads formation that attracts user-clicking, the probability is high for the campaign to result in a lower effectiveness and, thus, a higher cost if the landing page does not express a sufficient loading time. Additionally, indeed, one of the most fundamental aspects of paid search advertising campaigns and their effectiveness is the landing page loading speed time [59].

Regarding the involved speed metrics that PSI uses, it is noted that LAMs administrators could use this tool, firstly to identify which metrics impact most on the loading speed time of a webpage, and secondly, to follow the recommendations of "what to rectify in the code." These recommendations are connected with an indicator of seconds reduction. For example, some of the recommendations include eliminating render-blocking resources, serving images in next-generation formats (WebP or AVIF) CSS and JavaScript minifications, or reducing unused CSS or JavaScript elements. Google, as the PSI tool provider, suggests a code of the elements that could be optimized to reduce the webpage loading time. For the first contentful paint, one of the main issues that cause delays is the fonts loading time [60]. For the speed index, it is recommended to reduce JavaScript elements' execution time and to ensure that texts remain visible during webfont loading [61]. The time to interactive is also optimized when reducing the JavaScript execution time [62]. Several components were negatively impacted for the largest contentful paint, such as a slow server response time, the time needed to load up DOM resources, and client-side rendering [63]. The total blocking time indicates reducing the unnecessary or inefficient JavaScript elements [64]. Lastly, the provision of the size attributions in images or video elements to help browsers allocate the correct amount of space in potential requests and the prevention of inserting content above existing content within the same code elements are some of the most vital recommendations to reduce the cumulative layout shift [65].

*5.3. Theoretical Implications*

Recent reports indicate that all the requests to web servers for displaying webpages to users have continuously increased over the last five years since webpages' size are also increased [66]. This fact unintentionally impacts the websites loading speed time to increase as well. That is, the more content users need to retrieve, the more resources are needed. In this study, we shed light and develop a data-driven methodology to discover what metrics impact most to the reduction in the webpages' loading speed time. To discover this impact, a sample of 121 unique homepages of libraries, archives, and museums have been used as a sample while adopting PageSpeed Insights. One step forward, we explored in-depth PSI involving twelve metrics (six for mobile and six for desktop). To the best of our knowledge, there is not a prior research approach that involves the whole diversity of the available PSI indicators to understand, suggest, and prioritize through predictive model rectifications to reduce the loading time. Of course, there are prior approaches that tried to utilize PSI; however, with a limited amount of examined webpages while taking into consideration only the two major metrics, namely, the total PSI score for mobile and desktop [28–31].

In terms of the LAMs research context, very few approaches tried to understand libraries, archives, and museums' webpages' speed performance up to date [34,35]. Therefore, we believe that this effort adds novel research knowledge about libraries, archives, and museums in terms of their webpages' speed performance and how it could be explored and measured in-depth. Moreover, the results could also stand as an initial compass and summarization indicator regarding the speed performance of LAMs websites worldwide.

### 5.4. Future Implications

The more the research sample, the more stable the findings. In this respect, one of the first potential attempts is to increase the sample of the examined LAMs webpages. Of course, in this paper, we retrieved the speed performance of 121 homepages and compared it to prior approaches with a limited number of cases, and this number is at least higher to generalize the findings. However, a future step is to increase the number of cases involved to estimate the speed performance through this study's proposed reliable and valid measurement system. A step that will help even more to generalize the results in a broader context.

Another aspect that needs to be mentioned and explored furtherly is the determination of the weights that the PSI assigns in each of the six metrics on mobile and desktop. According to the Lighthouse scoring documentation of the last version [67], the first contentful paint receives a weight of 10% in the total scoring. Similarly, 10% of the weight is assigned to the speed index and time to interactive metrics. The largest contentful paint receives a weight of 25%, while a cumulative layout shift has a percentage of 15%. The highest weight is assigned to the metric total blocking time, receiving 30% as a PSI metric. According to the regression equation outputs of this study, the highest predicted change is observed in the first contentful paint for mobile versions (1.894). At the same time, the desktop received the highest predictive value at the largest contentful paint (5.776). Although these metrics extracted a higher predictive value of change in this study, the Lighthouse documentation assigned a 10% in the first contentful paint and 25% in the largest contentful paint metric. To this respect, further efforts are needed first to increase the sample size of this research. After that, to re-perform the webpages' speed performance estimations to compare the regression outputs with the pre-determined weights that Lighthouse 8 assigns to the PSI metrics.

Lastly, prior research approaches in the LAMs context tried to establish reliable and valid performance measurement systems, estimating the SEO, speed, and security conditions for the overall websites [6,41]. However, combining or comparing the prior proposed models—especially the website speed conditions—could be advantageous to this study's speed performance measurement system. This will cover both the micro levels (individual webpages) and the macro levels (websites as a whole) for identifying, retrieving, and analyzing the speed performance data and optimizing the LAMs websites' speed loading time.

**Author Contributions:** Conceptualization, C.X., F.-R.D., I.C.D. and D.K.; methodology, C.X., F.-R.D., I.C.D. and D.K.; formal analysis, C.X., F.-R.D., I.C.D. and D.K.; data curation, C.X., F.-R.D., I.C.D. and D.K.; writing—original draft preparation, C.X., F.-R.D., I.C.D. and D.K.; writing—review and editing C.X., F.-R.D., I.C.D. and D.K. All authors have read and agreed to the published version of the manuscript.

**Funding:** This research received no external funding.

**Institutional Review Board Statement:** Not applicable.

**Informed Consent Statement:** Not applicable.

**Data Availability Statement:** Dataset has been published at Zenodo research repository. Further information can be found at: https://doi.org/10.5281/zenodo.7266363.

**Acknowledgments:** Dedicated to the students of the Department of Archival, Library and Information Studies of the University of West Attica.

**Conflicts of Interest:** The authors declare no conflict of interest.

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
