# Peer review of "Speed Matters: What to Prioritize in Optimization for Faster Websites"

_2813-2203, doi:10.3390/analytics1020012_

Round 1

Reviewer 1 Report

The paper proposed establishment of methodological framework composed of three different stages to measure the speed performance of libraries, archives, and museums webpages.

Results of shown methodology could be helpful for libraries, archives, and museums administrators to set a data-driven framework of prioritization regarding the rectifications for optimized webpages loading speed time.

Number of references is 67, which is enough and advisable.

The paper is understandable, well structured and sufficiently detailed.

The theoretical and practical conclusions drawn are very relevant and important.

By my opinion, the article does not need to be corrected from point of view of researching aim.

My suggestions below, which seem essentially formal, will improve the clarity of the paper.

In general, I recommend the use of more consistent wording - using passive voice.

May suggestions to improve the quality of paper:

suggestion 1:

Please, depict structure of paper in separated paragraph (rows 68 – 74).

suggestion 2:

Please, do not use “In the following table (Table 2), we …” and similar phrasing (row258). Instead I suggest use “In the Table 2, we …” form .

suggestion 3:

Do not pigment Figure 1, because its text is not readable!!!

suggestion 4:

Do not use phrasing that can expresses authors’ uncertainty.

See:

“… we try to establish a methodological …” (row 58);

“… we tried to examine …” (row452).
You established, and you examined.

Author Response

Dear Reviewer, 

We really thank you for your comments as they gave to us the opportunity to improve the quality of our study. Below you can find the answers per comment. 

suggestion 1:

Please, depict structure of paper in separated paragraph (rows 68 – 74).

Answer: Yes. Thank you for that. We put the structure of the paper in a separate paragraph.

suggestion 2:

Please, do not use “In the following table (Table 2), we …” and similar phrasing (row258). Instead I suggest use “In the Table 2, we …” form .

 Answer: Yes. Thank you for that. We changed all the cases for the sake of users' better readability.

suggestion 3:

Do not pigment Figure 1, because its text is not readable!!!

Answer: Yes. The figure has been changed using lighter grey color to improve its readability.

suggestion 4:

Do not use phrasing that can expresses authors’ uncertainty.

See:

“… we try to establish a methodological …” (row 58);

“… we tried to examine …” (row452).
You established, and you examined.

Answer: Yes. Thank you for this comment. We changed all these sentences. For each one correction, a track change indicator has been used within the doc.  

Reviewer 2 Report

This paper proposes a framework to evaluate the metrics that affect the loading speed of websites of libraries, archives, and museums. This framework contains three parts, including data collection, metrics validity, and using a regression model to determine the impactive metrics. This paper is well-written and thoroughly researched, and the reference works are sufficiently cited and discussed. 

There are minor changes needed mostly on the grammar errors and formatting. In the abstract, the sentence in line 15 - 17 is not complete. Tables presented in the paper are not in a consistent style. Table 1 is hard to read.

Author Response

Dear Reviewer, 

We really thank you for your time and comments, as they gave to us the opportunity to furtherly improve the readability and structure of the paper. Below you can find the answers per comment. 

 _______________________

There are minor changes needed mostly on the grammar errors and formatting. In the abstract, the sentence in line 15 - 17 is not complete. Tables presented in the paper are not in a consistent style. Table 1 is hard to read.

Answer: We improved the readability of Table 1 using borderlines per row. We also used the proper style of the template per table regarding its fonts.

In the abstract, the sentence in line 15 - 17 is not complete. Answer: We thank you for this comment. It was our mistaken omission. We corrected the sentence.

Regarding the grammar errors: Thank you for that comment. Once more, we proofread each line carefully and proceeded with several grammar corrections. For each correction, the doc track changes have been used.